# Sculpting an Embryo: The Interplay between Mechanical Force and Cell Division

**DOI:** 10.3390/jdb10030037

**Published:** 2022-09-01

**Authors:** Nawseen Tarannum, Rohan Singh, Sarah Woolner

**Affiliations:** Wellcome Trust Centre for Cell-Matrix Research, Division of Cell Matrix Biology and Regenerative Medicine, School of Biological Sciences, Faculty of Biology, Medicine & Health, University of Manchester, Oxford Road, Manchester M13 9PT, UK

**Keywords:** mechanical force, biomechanics, cell division, mitosis, mitotic spindle, cell division rate, cell division orientation, cell shape, morphogenesis, embryogenesis

## Abstract

The journey from a single fertilised cell to a multicellular organism is, at the most fundamental level, orchestrated by mitotic cell divisions. Both the rate and the orientation of cell divisions are important in ensuring the proper development of an embryo. Simultaneous with cell proliferation, embryonic cells constantly experience a wide range of mechanical forces from their surrounding tissue environment. Cells must be able to read and respond correctly to these forces since they are known to affect a multitude of biological functions, including cell divisions. The interplay between the mechanical environment and cell divisions is particularly crucial during embryogenesis when tissues undergo dynamic changes in their shape, architecture, and overall organisation to generate functional tissues and organs. Here we review our current understanding of the cellular mechanisms by which mechanical force regulates cell division and place this knowledge within the context of embryogenesis and tissue morphogenesis.

## 1. Introduction

Embryogenesis is a highly coordinated process involving cell proliferation within a complex and dynamic mechanical environment. In recent years, significant progress has been made in our understanding of how mechanical forces regulate cell division in cells and tissues and the crucial role this mechanoregulation plays in the developing embryo [1,2,3,4,5,6]. Additionally, the complex non-linear nature of the link between forces and divisions has also become apparent with forces impacting divisions and vice versa. Therefore, embryonic cells are required to sense fluctuations in the force regime in their vicinity to divide accordingly while also inherently contributing to the mechanical landscape. Another key question that has come forth is how forces are sensed by mitotic cells. In this regard, the debate of direct force sensing versus force-sensing via changes to cell shape has received prominent attention. However, the molecular details underlying each of these mechanisms and whether one process dominates over the other or acts synergistically with the other to bolster robust mechanosensitive cell divisions remain areas of ongoing research. 

In this review, we discuss recent findings revealing how forces and cell divisions coordinate tissue morphogenesis during the development of different model organisms. Using a cells-to-tissues approach, we summarise how forces are generated and transmitted by cells and tissues and how force-mediated divisions impact tissue formation and function. We then focus on the cellular mechanisms underlying mechanosensitive cell division and outline key molecular players that may mediate cell divisions according to an external force.

## 2. Mechanical Force Generation and Transmission in Tissues

Individual cells within a developing embryo experience forces in the form of tension (stretching), compression (pushing), or shear (two forces acting in opposing directions) depending on the direction in which forces act on the cell [7]. These forces can arise extrinsically from the extracellular matrix (ECM), the tissue fluid, or as a result of morphogenetic movements. Cells can also generate forces intrinsically due to actomyosin contractility and the activity of molecular motors [8]. In epithelial tissues where cells are interconnected by cell-cell adhesions, forces generated in one cell can act on neighbouring cells via these adhesions (Figure 1 and Figure 2) [9].

Importantly, cell division itself can generate mechanical forces at different stages of the division process. Most animal cells round up as they enter mitosis due to detachment from the ECM substrate concomitant with actomyosin contractility and increased cortical stiffness [17]. During this process of mitotic rounding, cells generate both inward and outward extracellular forces on the surrounding cells. In squamous epithelia, mitotic rounding exerts inward forces pulling the neighbouring cells inwards, thereby stretching these cells [18]. In cuboidal or columnar epithelia, mitotic rounding generates outward forces that push on the neighbouring cells, thereby exerting compressive forces [19]. Studies on epithelial monolayers have shown that differential localisation of the actin-binding protein vinculin at cadherin-based cell-cell adhesions of the mitotic and neighbouring cells impacts mitotic rounding-mediated force generation [18]. At cadherin-based junctions shared between a mitotic cell and a cell at interphase, vinculin is recruited specifically from the non-dividing cell to maintain junction integrity. However, in the mitotic cell, vinculin is not recruited to the junctions to ensure proper mitotic rounding, thereby exerting tensile forces on the neighbouring cell [18]. As cells proceed through mitosis and commence division elongation at anaphase, they exert outward forces along the axis of division. These forces stem from a combination of interpolar microtubule elongation and contraction of the cytokinetic ring [20,21,22]. Outward force generation is also sustained throughout the cell spreading of the daughter cells at the end of mitosis [20].

## 3. Cell Division as a Foundation of Tissue Architecture

For embryogenesis, the rate and orientation of cell divisions must be precisely regulated in line with the mechanical environment of the developing tissue. The rate of division regulates the temporal mass of the tissue and an unregulated high division rate can contribute to the formation of tumours in the embryo [23]. 

Cell division orientation is dictated by the mitotic spindle which is dynamically positioned to determine the axis along which the cell will divide. The position of the spindle and the resulting division orientation has important consequences for the fate of the cell and for tissue organisation. The vast majority of cells in an epithelium undergo symmetric divisions parallel to the plane of the tissue, thereby generating identical daughter cells that help to expand the tissue along the epithelial plane. On a tissue scale, if these planar divisions orient in the same direction, tissue elongation can occur whereas if these divisions are randomly oriented, growth occurs in all directions. For example, during *Drosophila* germ band extension, divisions oriented along the axis of elongation are vital to driving the extent and rate of tissue elongation [24]. In addition to oriented divisions, modulation of cell proliferation can also play a role in tissue elongation. For example, during gastrulation in *Xenopus* embryos, inhibition of cell proliferation is required in the paraxial mesoderm to ensure complete elongation [25]. However, it has to be pointed out that sometimes divisions are not involved at all in tissue elongation: in the chick embryo, proliferation, or lack thereof, in the posterior presomitic mesoderm does not contribute to axis elongation [26]. Divisions can also be oriented perpendicular to the plane of the tissue, and in most cases, this leads to asymmetric divisions generating daughter cells with different cell fates while causing tissue stratification [7] (Figure 3). A combination of these directional division orientations is pivotal to tissue morphogenetic events such as gastrulation [27], formation of the neural tube [28,29], and shaping of organs [30], with failure to orient divisions appropriately contributing to abnormalities during morphogenesis and organogenesis [29,30].

Crucially, mechanical forces impact the choice between planar and out-of-plane divisions. In suspended epithelial monolayers, compression forces lead to out-of-plane divisions whereas stretch leads to more planar divisions [31]. This was also shown in mouse embryos that are defective for the planar cell polarity gene, *Vangl2*, and are unable to undergo neural tube closure. In these embryos, increased cell crowding within the basal layer of the epithelium leads to thinner and taller cells that then undergo out-of-plane divisions. Upon application of mechanical stretch along the epidermal plane, these out-of-plane divisions switch to predominantly planar divisions [32]. Concerning the role of forces on asymmetric cell divisions, most studies have focused on intracellular forces with few studies reporting the direct role of external forces on asymmetric divisions [33]. Intriguingly, one such study on keratinocytes has shown that mechanical stretch orients asymmetric divisions in these cells [34]. However, since the majority of studies investigating mechanosensitive cell divisions have concentrated on symmetric divisions, these will be the focus of this review. Mechanical forces are known to impact both the rate and orientation of cell divisions [1,4,5,35,36,37,38,39,40,41,42,43,44,45,46] and, therefore, play a critical role in orchestrating tissue form and function via robust control of cell divisions.

## 4. Forces and Cell Division Rate

Generally, mechanical tension on a tissue leads to an increased rate of cell division. Studies on monolayers and tissues have shown that the application of an external stretch triggers cell cycle progression and entry into mitosis to help restore the homeostatic tissue stress [35,43,47,48]. For example, in zebrafish embryos undergoing epiboly, cells in the enveloping cell layer undergo rapid cell divisions under tension and this is important for tissue morphogenesis [1]. 

Studies on epithelial monolayers have shown that mechanosensitive cell proliferation can be affected by differential spatial constraints. Spatial constraints create regions of high and low cell densities, and cells in each region respond differently to forces. In regions of relatively low cell density, cells are under stretch and tend to proliferate faster whereas cells in more crowded regions are under compression and undergo extrusion from the monolayer [39,49]. 

In response to constraint, the cell division rate is controlled by mechanosensitive checkpoints at different stages of the cell cycle. Studies on cell spheroids confined within viscoelastic hydrogels with different stress relaxation properties have shown that in slow-relaxing gels, cell cycle progression is inhibited at the G0/G1 boundary due to confinement. In fast-relaxing gels, however, cells can transition into G1 phase and ultimately into S phase [50]. The G1/S checkpoint can also prevent cells from entering S phase if sufficient space is unavailable due to cell crowding. Application of external stretch on a monolayer and the consequent reduction in crowding induces the transition of G0/G1 cells into S phase whereas compression prevents it [48]. Similarly, a bidirectional external stretch of quiescent epithelial monolayers induces rapid cell cycle re-entry followed by progression into S phase [35]. In contrast, biaxial planar compression of epithelial cells inhibits cell cycle progression through arrest at S phase [51]. Changes in cell numbers within epithelial monolayers can also be sensed at the G2/M boundary to regulate mechanoresponsive proliferation. In a dense epithelium, cells are maintained at G2 phase but an increase in mechanical tension, either upon uniaxial stretch or epithelial expansion upon wounding, triggers cell entry into mitosis [47]. Overall, mechanosensitive cell divisions, triggered by cell cycle checkpoints, may help to maintain optimal cell density in the overall tissue with constant feedback from the physical environment. 

This feedback mechanism is important in the *Drosophila* wing imaginal disc where differential growth rates generate mechanical stress. In this tissue, high proliferation rates relieve tension at the cell-cell junctions and function as a cue to then lower the growth rate, thereby modulating tissue homeostasis [52]. A similar homeostatic role has also been indicated by studies on *Xenopus* embryonic explants where there are temporal fluctuations in cell proliferation rates under mechanical stretch. Upon uniaxial stretch, there is an initial phase of high proliferation rate followed by a return to steady-state rate after 40–60 min of stretch application. Furthermore, the individual cells that divide are more likely to be under tension than under compression [43]. This is suggestive of a stress-relief mechanism by which added mechanical stress is dissipated by a peak in the division rate until the tissue reaches an equilibrium state, at which point, division rates decrease to keep up with the steady-state stress.

Biochemical signals may also differentially influence cell proliferation leading to alterations in tissue mechanics during organogenesis. In the developing chick embryo, BMP signalling-dependent differential proliferation of the gut tissue modulates compressive forces on the gut as it elongates against the dorsal mesentery, an elastic membranous tissue that anchors the gut to the abdominal wall. These compressive forces facilitate the looping of the gut to form the digestive tract [3]. 

Cell proliferation in response to forces can also be influenced by cell and tissue geometry. In three-dimensional epithelial acini, cells experience greater tension compared to those in monolayers due to increased hydrostatic pressure within the acinar lumen, and thereby have high cell proliferation rates [53]. In the *Drosophila* ovarian follicular epithelium, the growth of the egg chamber exerts mechanical strain on a group of cells leading to the flattening of these cells. As a result, the apical surface of the flattened cells experiences a stretch and ultimately triggers signalling cascades to induce subsequent cell proliferation [2].

In addition to studies where forces were applied to cells or tissues, the application of external forces directly on the nucleus using atomic force microscopy has been shown to induce changes in the cell cycle [54]. During the G1/S transition, the nuclear envelope undergoes flattening. Loss of myosin-2 activity prevents nuclear flattening and stops entry into S phase. Upon exertion of compressive forces on the nucleus in myosin-2-deficient cells, this effect was alleviated and cells were able to transition into S phase. Therefore, mechanical forces can impact the shape of the nucleus to subsequently affect divisions [54]. 

## 5. Forces and Cell Division Orientation

One of the long-standing theories describing cell division orientation is Hertwig’s rule which states that cells orient their spindle along their longest axis of shape [55]. However, cells also seem to align divisions with the axis of greatest tensile stress [1,4,36,38,40]. Whether this effect is due to force itself or associated cell shape changes has been an ongoing debate in the field since the application of force changes the shape of cells. 

Pioneering work investigating division orientation with cell geometry implemented microfabricated chambers to “force” cells to adopt the shape of the chamber. When sea urchin eggs were encased in chambers of different shapes, spindles mostly aligned with the long axis. However, in some shapes, the spindle aligned with the axis of symmetry rather than the long axis. In these cells, spindle alignment was set by the position of the nucleus at interphase or early prophase. These shape exceptions suggested a model whereby centrosomal microtubules spread out to recognise cell geometry and exert forces in scale with their length to orient the spindle with the shape of the cell (Figure 4A) [42]. In addition to cell geometry, attachment to the ECM is an important cue in mediating spindle orientation with cell shape. When cells were plated on ECM micropatterns of different shapes, they aligned their spindles along the longest axis of interphase shape. This visualisation of Hertwig’s rule is a result of the segregation of cortical cues to ECM-adhered regions at interphase. As the cells undergo mitotic rounding, retraction fibres attach the cell to the ECM and maintain the localisation of these cortical cues, which then interact with spindle poles via astral microtubules to dictate spindle alignment (Figure 4B) [44,45]. Further studies using laser ablation to “cut” some of the retraction fibres showed that the spindle aligned with the shape defined by the remaining retraction fibres, indicating that these fibres provide extrinsic cues to align the spindle (Figure 4C). The involvement of external force in orienting the spindle was more directly shown when cells subjected to an external stretch aligned their spindles along the stretch axis [37].

Following these seminal works, further research has been suggestive of the consensus that cells in monolayers or tissues sense forces via changes in cell shape to orient the spindle. However, a growing body of evidence is also counteracting that force can orient divisions regardless of cell shape. Interestingly, there is also a possibility that both mechanisms may function in parallel to accurately position the spindle. In essence, the mechanisms identified to date are context-dependent and a generalised mechanism may be unlikely to capture the robustness of the process.

### 5.1. Cell Shape Change as the Primary Cue in Mechanosensitive Cell Division Orientation

Experiments, where tissue tension can be precisely controlled by the application of external forces, have indicated that cell shape seems to be a predominant factor in orienting cell divisions rather than force. Stretching suspended epithelial monolayers leads to a global bias in the cells’ long axis of shape orientation along the direction of stretch. Although these cells divide along the stretch axis, cells with their long axis oriented perpendicular to the stretch axis tend to divide with their shape rather than the stretch [46]. A cell shape preference for division orientation was also shown in stretched *Xenopus* embryonic epithelial tissue explants where mathematical modelling was used to infer mechanical stress in individual cells. In this tissue, spindle orientation is best predicted by the cell shape axis which is dictated by the position of the tricellular vertices (TCVs), where at least three cells meet. The TCV-defined shape axis aligns perfectly with the direction of local tensile stress but in the *Xenopus* explant, it is the level of cell elongation rather than mechanical stress intensity which most strongly determines spindle orientation [43].

Force sensing via cell geometry has also been shown in the context of developing tissues where mechanical forces arise intrinsically. During the development of the *Drosophila* wing imaginal disc, most divisions in the central wing pouch are oriented along the proximal-distal (P-D) axis of the epithelium. However, at the tissue boundaries, cells orient their division perpendicular to the P-D axis in response to global mechanical force. This force originates from differential cell division rates within the tissue and influences cell shape leading to spindle alignment. Additionally, overgrowth of cells within a region generates stretch forces which act on neighbouring cells, inducing spindle re-reorientation in the neighbours [41].

Force-driven changes in cell shape are key to orienting divisions for polarised tissue growth during morphogenesis. In the *Drosophila* wing imaginal disc, cells at the edge of the tissue are under tension while those in the centre are under compression. This affects cell shape within the wing pouch and cells in the periphery respond to this stress by polarising cortical actomyosin. This polarisation eventually orients spindles along the stretch axis, thereby leading to directional tissue growth [56]. Additionally, forces and the consequent changes to cell shape ensure that directional tissue growth is tightly controlled by feedback from cells themselves. In zebrafish embryos during epiboly, cells in the enveloping cell layer undergo cell-shape oriented divisions along the tension axis to relieve anisotropic stress. However, this is a reciprocal process where the degree of anisotropic tissue tension depends on the ability of cells to orient their divisions while division orientation also relies on overall tension [1]. 

Mechanosensitive, cell-shape-driven, division alignment is also important for organogenesis. During the development of the airway epithelium in mice, overall cell division orientation in the tissue is governed by two distinct behaviours of the spindle characterised as either “fixed” or “rotating”, determined by the shape of the cells. Cells with fixed spindles are more elongated, aligning divisions with interphase shape and their division orientation is established as soon as the cell enters metaphase. However, cells with rotating spindles are comparatively less elongated and align their divisions during metaphase. Crucially, mechanical forces act as the key regulator in maintaining a stable ratio between each spindle type [5].

When investigating the role of cell shape in spindle orientation, it is important to consider that most animal cells tend to round up as they enter mitosis [57]. Therefore, the question of how cell shape is sensed when cells undergo mitotic rounding becomes key to understanding how interphase cell shape dictates spindle alignment in rounded mitotic cells. This has been investigated in the *Drosophila* pupal notum epithelium where spindle orientation is defined by the interphase cell shape axis based on the position of the tricellular junctions (TCJs). As the cell transitions from interphase to mitosis, the TCJs act as spatial cues to deliver interphase cell shape information to the mitotic cell to orient the spindle. Simulations and mathematical modelling showed that TCJ angular bipolarity is a better predictor of division orientation than metaphase cell shape and, in simulations, this TCJ bipolarity aligned with the axis of global mechanical strain [58]. Another important aspect to consider is how extrinsic forces from surrounding cells in a tissue coordinate with cell-intrinsic forces as a result of mitotic rounding to influence cell shape and subsequently mediate spindle alignment. This was investigated in early ascidian embryos where polarised cells undergo mitotic rounding and apical relaxation. Due to this reduction in the apical cortical tension, the apical surface area increases and the apex becomes more sensitive to shape changes due to external forces from neighbouring non-dividing cells. Ultimately, changes in cell shape as a result of the extrinsic force lead to division alignment with the force axis [38].

### 5.2. Force Can Be Sensed Independently of Cell Shape to Orient Cell Divisions

Although there is a clear role for cell shape in aligning cell divisions with an axis of global strain, evidence is growing that, in some systems at least, direct-force sensing, independent of cell shape, is crucial for division orientation. For example, studies on epithelial monolayers subjected to a low level of an external stretch have shown that cells orient their spindles along the stretch axis, irrespective of their long axis of shape [40]. Additionally, in suspended monolayers where compression leads to divisions outside the tissue plane but stretch leads to more in-plane divisions, these divisions occur independently of cell shape [31]. This was also observed in developing tissues in vivo. In the *Drosophila* ovarian follicular epithelium during the expansion of the egg chamber, the epithelium is under global tension along the elongation axis. Cell divisions in this tissue are oriented along the planar tissue axis predominantly along the direction of tissue elongation. Interestingly, the cell shape orientation does not align in the direction of tissue expansion suggesting that divisions in this tissue do not orient with cell shape [36]. Furthermore, in the *Drosophila* embryo during segmentation, actomyosin supracellular cables formed at the parasegmental boundaries exert tension on the boundary cells and orient their division perpendicular to the boundaries. These cells orient mitotic spindles along the stress axis and not along the cell shape axis. Additionally, upon laser ablation of the supracellular cables, the spindles rotated closer to the long axis of cell shape indicating that actomyosin-derived tension is the more important factor over cell shape in orienting divisions in this tissue [4]. A more recent study on *Drosophila* embryos during gastrulation has shown that divisions in the mitotic domains within the dorsal head orient according to morphogenetic forces arising from the invagination of the mesoderm in a manner that is independent of cell shape [59]. Also recently, using optogenetics to induce inhomogeneous cortical tension on quasi-spherical mitotic cells, it was shown that in the regions of high tension, there is a decrease in the pulling of cortical force generators on astral microtubules. This leads to spindle rotation, and subsequent division orientation, away from the tensed cortical domains. No correlation was found between the final spindle rotation angle and cell shape, indicating the possibility that force may be sensed directly in this context [60].

Complimentary to the role of direct force sensing in division orientation, the uncoupling of cell shape from force sensing has also been demonstrated in regards to division rate in *Xenopus* animal cap tissues under stretch when myosin-2 is depleted. The cells that divide tend to be under net tension and, in myosin-2-depleted tissues, dividing cells are similar in size to their non-dividing contemporaries (compared to dividing cells being the largest cells in control tissues), thereby uncoupling the contribution of cell shape changes from the force. This indicates that, in the context of cell proliferation, mechanical stress can be sensed directly without any contribution from cell shape [43]. 

### 5.3. Force and Cell Shape Function Synergistically to Orient Global Cell Divisions in Tissues

A recent study has indicated that within the same tissue, cells may be able to respond to both cell shape and force differentially. During axis elongation in *Drosophila* embryos, mesectoderm cells within the ventral germband divide along the anterior-posterior (A-P) axis due to tissue tension along this axis. These tension-oriented divisions help to reduce tension on the tissue while facilitating tissue elongation. However, the lateral cells in this tissue undergo divisions that are oriented more uniformly. Interestingly, in the mesectoderm cells, the axis of divisions deviated significantly from interphase cell shape whereas in the lateral cells, divisions were more faithful to cell shape. Therefore, this suggests that the mesectoderm cells may be responding to directional cues compared to shape-determined cues in the lateral cells [6].

## 6. Mitotic Rounding as an Additional Cue for Morphogenesis

Aside from the end result of mechanosensitive cell divisions in shaping a tissue, mitotic rounding within the process of cell division itself may contribute to tissue morphogenesis. Since extensive work has focused on mitotic rounding, in this review we discuss some key examples and interested readers are thereby referred to an excellent review for more in-depth information [17]. During zebrafish inner ear morphogenesis, mitotic rounding of cells facing the epithelial lumen exerts pulling forces on the luminal surface of the epithelium, thereby facilitating lumen expansion [61]. In mouse embryos during the formation of intestinal villi, mitotic rounding in the compressed regions of the epithelium facilitates rapid invagination of the epithelium to demarcate the villi [62]. Furthermore, during the development of the *Drosophila* tracheal system, mitotic rounding promotes epithelial invagination of the tracheal placode. During the initial slow phase in this process, apical constriction of central cells within the placode initiates the formation of a pit. This is followed by a faster phase during which mitotic rounding of the central cells accelerates the deepening of this pit, thereby buckling the epithelium in a process that involves actomyosin contractility in the surrounding cells [63]. In zebrafish embryos, blastoderm spreading requires rapid fluidisation of the tissue which is facilitated by the destabilisation of cell-cell contacts upon mitotic rounding [64]. Additionally, in mouse embryos during gastrulation, mitotic rounding is one of the contributing factors in epithelial cell rearrangements leading to epithelial-mesenchymal transition in the posterior epiblast [65].

## 7. Potential Mechanotransducers Linking Forces to Mitosis

### 7.1. Mechanosensitive Channels

Recent work has indicated that the stretch-sensitive calcium ion channel Piezo1 is required to regulate cell division rate according to force. In monolayers with regions of different cell densities, Piezo1 localises differentially within cells in each region. In the sparse regions where cells are under stretch, Piezo1 localises to the cell membrane and cytosol where it is activated to allow calcium influx into the cell. This leads to calcium-dependent phosphorylation of ERK1/2 via MEK1/2 and subsequent cyclin B transcription, thereby triggering cell entry into mitosis (Figure 5) [39]. However, within cells in denser regions, Piezo1 forms aggregates in the cytosol and these cells are more likely to undergo extrusion [39,49]. Another mechanosensitive calcium ion channel, transient receptor potential vanilloid-type 4 (TRPV4), has been shown to function in cell spheroids to trigger cell cycle progression under low confinement conditions. In slow-relaxing viscoelastic gels, the cell cycle is halted at the G0/G1 boundary due to confinement. However, in fast-relaxing gels, cells transition into G1 phase and this activates TRPV4. This leads to downstream activation of the PI3K pathway and consequent inhibition of p27, thereby triggering entry into S phase (Figure 5) [50].

### 7.2. Cadherin

Cell-cell junctions are known to be important in sensing and transducing force. However, they have also been implicated in spindle orientation according to force. Studies on suspended monolayers have shown that a sufficient amount of junctional tension is required to ensure the alignment of planar divisions with force [31]. Among junctional components, cadherins have been the most widely implicated candidates in force-sensitive cell divisions. In quiescent epithelial monolayers under stretch, E-cadherin acts as a signalling centre and causes the nuclear localisation and transcriptional activation of the growth-regulator, yes-associated protein (YAP), and β-catenin. Transcriptional activation of YAP is transient and enables cell cycle re-entry from quiescence. β-catenin is activated afterwards, independently of YAP, over a longer timescale and promotes entry into S phase [35]. For cells to then enter mitosis, a combination of mechanical stretch-mediated phosphorylation of β-catenin and its stabilisation by Wnt signalling is required (Figure 5) [66]. E-cadherin also senses force differentially between cell-dense and sparse regions to influence cell division rate. In a dense epithelium where forces on E-cadherin are reduced, the G2/M checkpoint regulator Wee1 inhibits CDK1 and maintains cells at G2 phase. However, increased mechanical tension, either upon uniaxial stretch or epithelial expansion upon wounding, is sensed by E-cadherin resulting in rapid degradation of Wee1. As a result, CDK1 is activated, triggering cell entry into mitosis (Figure 5) [47]. 

Differential force sensing by E-cadherin has also been shown in the *Drosophila* wing imaginal disc. In this tissue, cells that proliferate faster have reduced cytoskeletal tension along the cell-cell junctions. Under reduced tension at the E-cadherin junctions, Hippo signalling is upregulated by a decrease in the junctional localisation of Ajuba LIM protein (Jub) and Warts kinase (Wts). This ultimately results in reduced activity of Yorkie (Yki), the *Drosophila* homologue of YAP, leading to a decrease in cell proliferation [52]. Cadherins may also be a candidate for modulating force-dependent cell division in the *Drosophila* ovarian follicular epithelium where the growth of the egg chamber exerts mechanical strain, leading to cell flattening. Stretching of the apical surface reduces the localisation of Hippo pathway proteins Crumbs, Expanded, Kibra, and Merlin at the apical membrane. This downregulates Hippo signalling and induces the nuclear localisation of Yki to induce gene transcription and subsequent cell proliferation [2]. Interestingly, the atypical Fat proteins, which are structurally and functionally distinct from classical cadherins like E-cadherin [67,68], may be involved in force-dependent cell divisions. Fat localises to the apical membrane in *Drosophila* epithelia and may interact with Expanded and Merlin [69,70], thereby hinting at the possibility that Fat may be able to transduce forces to control cell proliferation via the Hippo signalling pathway.

### 7.3. LGN/NuMA

Recent studies have highlighted the mechanoresponsive potential of proteins closely associated with the mitotic spindle in facilitating spindle alignment and subsequent division orientation. Two such candidates are Leu-Gly-Asn-enriched protein (LGN) and its binding partner nuclear mitotic apparatus protein (NuMA) both of which are core proteins of the spindle orientation machinery (Figure 6) [71,72]. Additionally, E-cadherin binds to LGN via its cytosolic domain [73], thereby enriching cortical force generators in the vicinity of cadherin-based adhesions.

Experiments on epithelial cell monolayers have shown that spindle orientation to an externally applied stretch depends on E-cadherin and LGN. Upon mechanical stretch, an increase in tension at E-cadherin junctions recruits LGN to the junctions, thereby polarising LGN localisation at the cortex (Figure 7) [40]. Similarly, in stretched *Xenopus* embryonic tissue explants, where spindle orientation aligns with the cell shape as defined by the position of TCVs, the co-alignment of spindle orientation with cell shape involves C-cadherin and LGN [43]. Additionally, in *Drosophila* embryos during gastrulation, cells within mitotic domains in the dorsal head experience morphogenetic forces via adherens junctions which leads to polarised localisation of Pins (the *Drosophila* homologue of LGN) and ultimately cell division orientation with the force [59].

Furthermore, the role of NuMA in mechanosensitive spindle orientation was highlighted by experiments on cultured keratinocytes subjected to external stretch. The stretched cells align their spindles with NuMA crescents at the cell cortex suggesting that cortical NuMA may be important in stretch-induced spindle alignment (Figure 7). Moreover, knockdown of NuMA in these cells disrupts spindle alignment with the stretch axis [34]. A key role of NuMA in this process was further indicated by studies on the *Drosophila* pupal notum epithelium. In this tissue, Mud, the *Drosophila* homologue of NuMA, localises to the TCJs. This localisation is observed during interphase and throughout mitosis even though the cells round up. Simulating mechanical stress on the tissue also highlighted that cell shape, defined by TCJs and localised Mud, aligns with the stress axis. Therefore, Mud localisation at the TCJs acts as spatial cues to deliver cell shape information from interphase to orient the spindle in rounded cells during mitosis [58]. However, it is important to note that NuMA and Mud have key differences in their protein domain structure and consequent subcellular localisation. Whereas Mud localises to the cortex throughout the cell cycle, NuMA is restricted to the nucleus at interphase and only translocates to the cortex during mitosis [75,76]. Therefore, it remains to be seen whether NuMA follows a similar localisation to Mud at the TCJs and whether mechanisms so far mostly investigated in *Drosophila* tissue apply in the context of vertebrate epithelia.

### 7.4. Actin-Binding Proteins

The actin-binding protein Canoe (the *Drosophila* homologue of Afadin), functioning as an adapter between the actomyosin cytoskeleton and cadherin-based junctions [77], has been implicated in orienting the spindle with force. During elongation of the egg chamber in the *Drosophila* ovarian follicular epithelium, cell division orientations in different axes were regulated by two distinct mechanisms. Whereas apical-basal orientation was guided by Pins and Mud, planar division orientation was dependent on apical tension sensing by the actin-binding protein Canoe [36]. Additionally, studies on stretching keratinocytes have suggested that the 4.1 family of actin-binding proteins (4.1R, 4.1N, 4.1G, and 4.1B) may interact with NuMA to orient mitotic spindles with external force (Figure 7) [34]. 

### 7.5. Myosin

Cortical actomyosin is important for precise spindle positioning [78,79]. Furthermore, myosin is involved in modulating both the rate and orientation of force-mediated cell divisions. In stretched *Xenopus* embryonic animal cap explants, cell proliferation is directly regulated by force in a myosin-2-dependent manner. As such, loss of myosin-2 function leads to reduced contractility and a sharp reduction in proliferation rate, almost ceasing in unstretched tissue. However, the application of external stretch rapidly increases the cell proliferation rate in myosin-2-deficient tissues suggesting that myosin-2 is required for cell proliferation in unstretched tissue but this requirement can be partially bypassed by the application of an external tissue stretch [43]. 

The importance of myosin-2 in orienting divisions with force was shown in zebrafish embryos undergoing epiboly. During this process, cells in the enveloping cell layer require myosin-2 activity to divide along the direction of tissue tension. In this tissue, loss of myosin 2 activity leads to a reduction in cell elongation and spindle positioning along the resultant longest axis of cell shape [1]. Furthermore, myosin-2 is also important during segmentation in *Drosophila* embryos where cells at the parasegmental boundaries experience tension from actomyosin supracellular cables that form at the boundaries. Myosin-2 in the supracellular cables orients the spindle with tension by capturing the centrosome closest to the boundary and restricting its movement. Therefore, local anisotropy in tissue tension generated by myosin-2 activity can orient divisions in vivo [4].

Intriguingly, recent studies on pulsating contractions during embryonic development have indicated a unique architectural form of actomyosin namely, pulsatile actomyosin that produces rapid pulses of force [80,81]. Actomyosin pulsing is a dynamic process in which F-actin and myosin-2 first assemble in a subcellular structure and then disassemble, on a time scale of minutes or less [82]. The resulting pulsatile mechanical forces have been shown to be involved in many morphogenetic events such as apical constriction, gastrulation, and dorsal closure in *Drosophila*, endodermal internalisation and polarisation of the zygote of *C. elegans*, and convergent extension movements in *Xenopus* [83,84]. Moreover, cortical actomyosin pulses have been observed during mitosis where they ensure the stability of cleavage furrow formation independently of the mitotic spindle [85]. To date, the majority of studies investigating the mechanical regulation of cell division have focused on large global changes in mechanical stress. Moving forward, it will be important to explore the impact of these fast, local stresses on proliferation in developing tissues.

## 8. Future Directions and Conclusions

During embryogenesis, developing tissues function as mechanosensitive entities to navigate through changes in the mechanics of their physical environment. By coordinating cell divisions according to forces, tissues not only respond to mechanical changes in their vicinity in the short term but also refine tissue mechanics to exert more global, long-term effects in shaping the embryo. 

In this regard, the effect of mechanical forces on cell division rate and orientation has been centre-stage in our understanding of how tissues respond to mechanical inputs. A key role of cell shape in regulating force-sensitive divisions has been established [1,5,38,41,43,46,56,58]. However, this seems context-dependent with a growing body of evidence suggesting that forces may also be sensed independently of cell shape [4,31,36,40,60]. Intriguingly, the concept of both mechanisms functioning within the same tissue is starting to emerge [6]. However, most studies have been restricted to investigating a single static force regime whether it be tissue stretch or compression. Therefore, it will be important to investigate how cells adapt to forces applied incrementally over time versus a fast, static application of force. This would provide key insight into understanding how divisions occur in vivo where the embryo likely experiences a combination of forces slowly building up over time and short-term faster impacts. 

Furthermore, one of the complex questions going forward is—how do cells react to a combination of external forces including shear strain? Since force and cell shape are intricately coupled, it would also be important to understand how cell shape is impacted by multidirectional forces to ultimately affect cell divisions. Another key aspect to consider is how intracellular forces generated by actomyosin contractility or the viscoelastic cytosol work alongside external forces to regulate mitosis. Whether there is an overlap of the key players in sensing internal and external forces to coordinate divisions is an interesting avenue to explore. 

With the concerted application of mathematical modelling and biology, we are starting to understand how individual cells in a tissue experience global mechanical force to undergo divisions. Ultimately, it is expected that we will be able to investigate this in whole embryos from the perspective of both global forces as well as forces within the immediate locality. This spatiotemporal approach to understanding mechanoresponsive cell divisions will refine our knowledge of how tissue mechanics is modulated at different stages of embryonic development. Taken together, this will serve as a road map toward unravelling the complexities underlying how a proliferating single cell is sculpted into a functional organism.

## Figures and Tables

**Figure 1 jdb-10-00037-f001:**
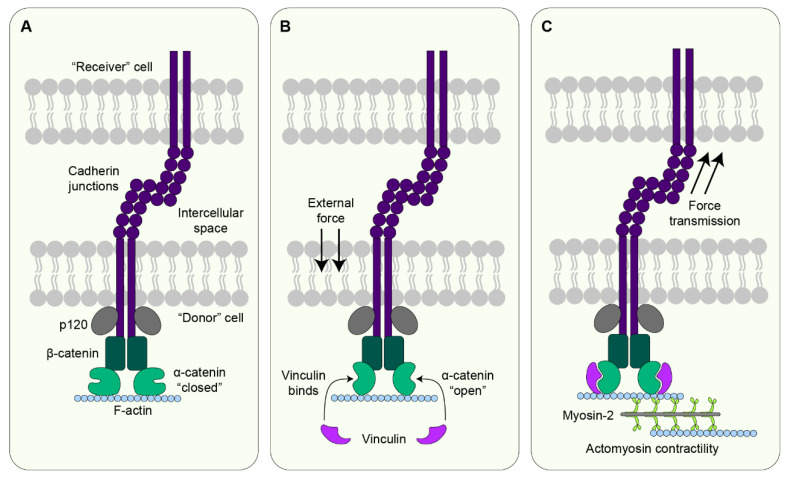
**Mechanical force transduction at adherens junctions.** Forces are transmitted through adherens junctions by cadherin dimers. The cytosolic tail of cadherin binds to p120 and β-catenin while the extracellular domain crosslinks with cadherin from the neighbouring cell. (**A**) In the absence of an external force, β-catenin is bound by α-catenin in its “closed” conformation to form the α-catenin-β-catenin-cadherin complex. In the “closed” state the vinculin-binding site on α-catenin is not accessible. (**B**) When the cell experiences a force, α-catenin undergoes a conformation change into the “open” state, thereby unmasking the vinculin-binding site and leading to vinculin recruitment at the junctions [10,11]. (**C**) Following vinculin recruitment, intracellular forces are generated by actomyosin contractility due to the movement of myosin head domains on F-actin filaments [12,13]. The cell that generates contractile forces acts as a force donor, and force is transmitted to the receiver cell via the junctions, thereby exerting pulling forces on the receiver cell. In addition to acting as a downstream responder to external forces, actomyosin contractility itself generates forces which act on cadherin junctions and elicit cadherin mechanotransduction in the receiver cell.

**Figure 2 jdb-10-00037-f002:**
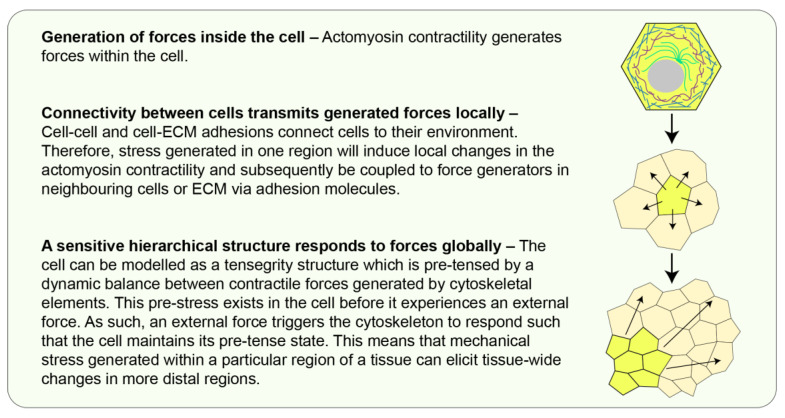
**Building forces from cells to tissues.** Forces generated within a cell can be transmitted to its interconnected neighbours. Cells can be modelled as tensegrity structures and tissues are, therefore, a collection of tensegrity structures which are inherently mechanosensitive [14,15,16].

**Figure 3 jdb-10-00037-f003:**
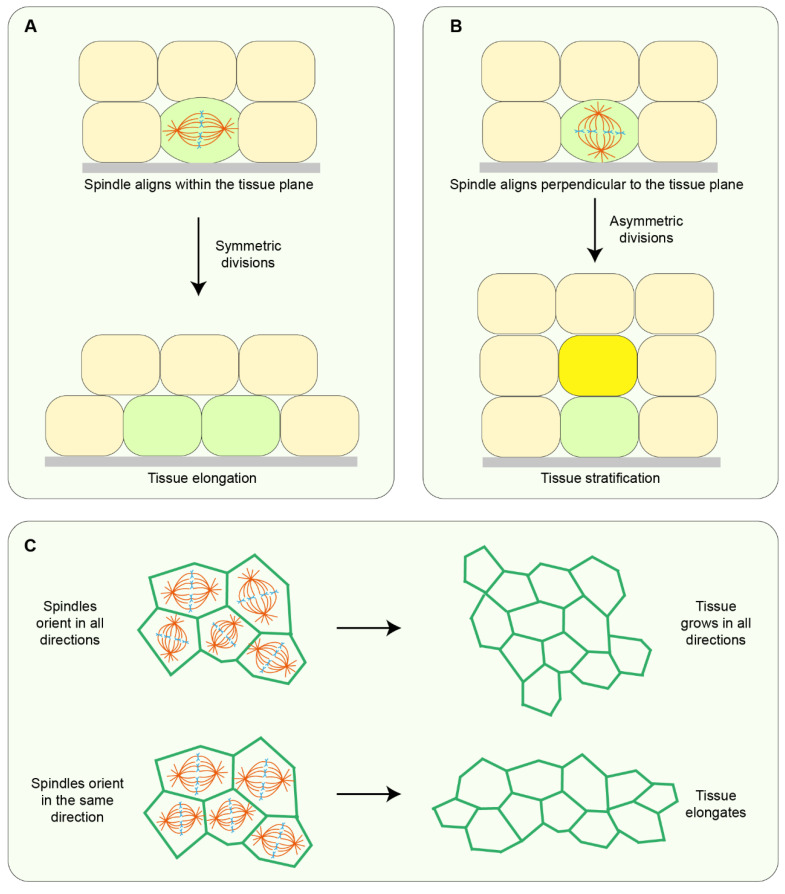
**The effect of cell divisions on tissue organisation.** (**A**) In symmetric divisions, spindle alignment parallel to the plane of the tissue generates identical, side-by-side daughter cells. (**B**) In asymmetric divisions, the spindle aligns perpendicular to the tissue plane and divisions generate daughter cells with different cell fates, which aid tissue stratification. (**C**) Symmetric divisions that orient uniformly within the tissue plane lead to tissue growth in all directions. However, if the spindles align in the same direction, tissue elongation occurs along the global axis of divisions.

**Figure 4 jdb-10-00037-f004:**
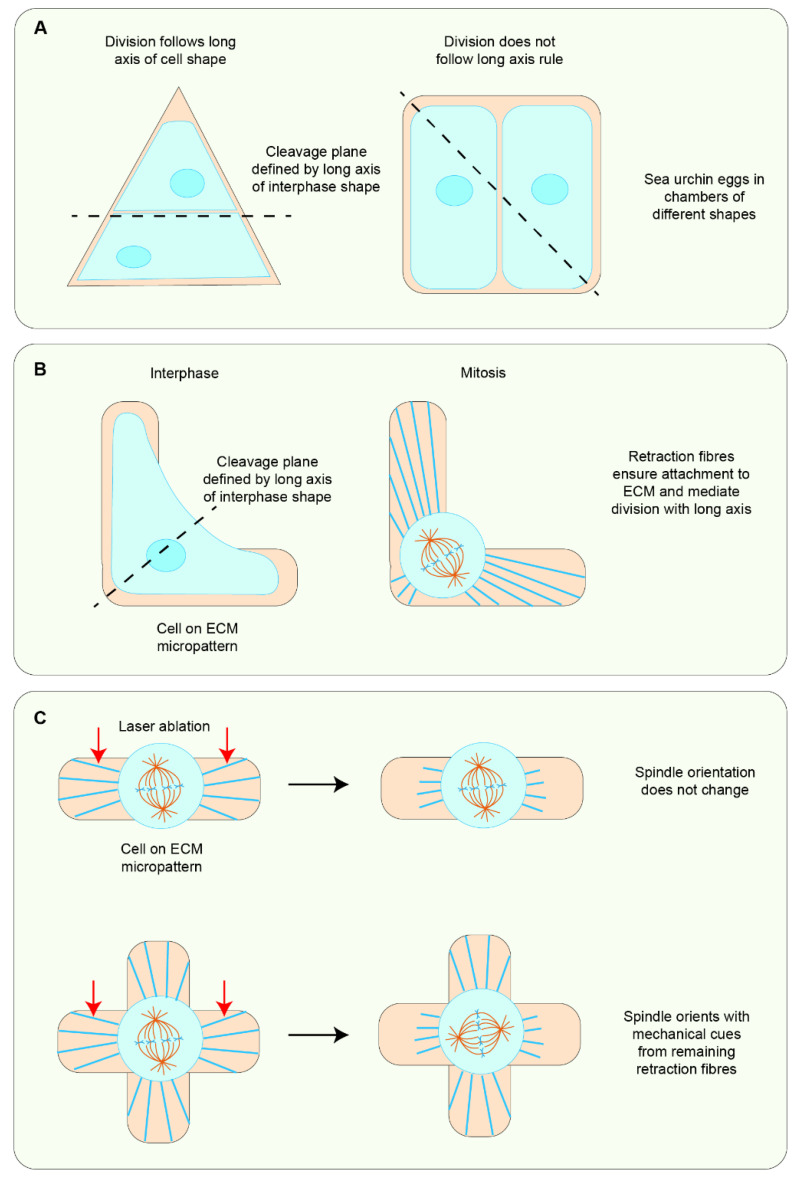
**Contribution of cell shape and force in spindle orientation.** (**A**) Cells placed in a microfabricated chamber assume the shape of the chamber but do not always follow the long axis rule of division. (**B**) Attachment of cells to the ECM via retraction fibres conserves the placement of cortical cues enabling division with the long axis of shape in rounded mitotic cells. (**C**) Using laser ablation to cut some retraction fibres leads to spindle alignment with cortical cues defined by the remaining retraction fibres.

**Figure 5 jdb-10-00037-f005:**
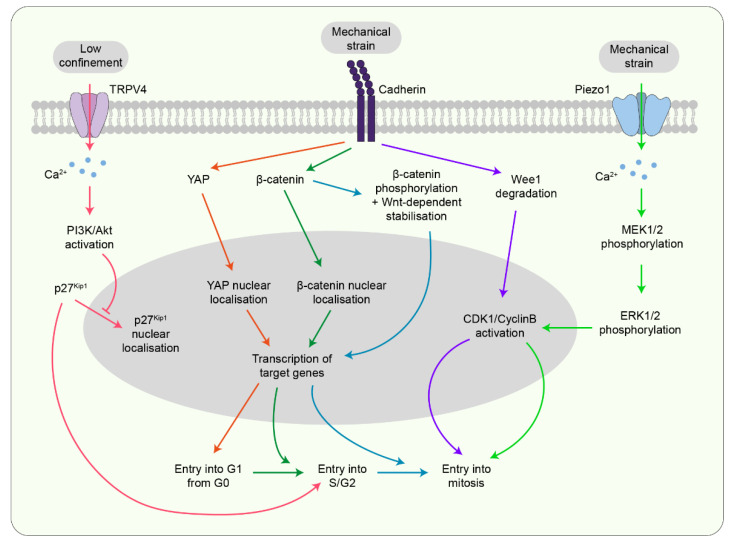
**Transmembrane proteins sense mechanical forces to influence cell proliferation.** Mechanosensitive channels such as Piezo1 and TRPV4 as well as cadherins act as signalling centres to influence cell divisions via control of the cell cycle at different checkpoints. Coloured arrows indicate different pathways.

**Figure 6 jdb-10-00037-f006:**
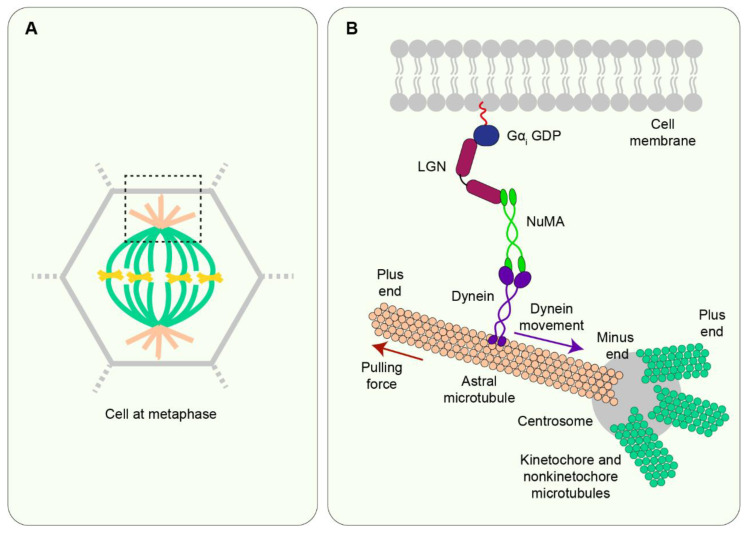
**The spindle orientation machinery in vertebrates.** (**A**) A schematic of a cell at metaphase is shown. The region selected by the black box is enlarged in B. (**B**) At interphase, NuMA is restricted within the nucleus but at the onset of mitosis, NuMA is carried to the minus end of microtubules (near the centrosomes) by dynein. Near this region, NuMA binds to LGN which, in turn, binds to Gα_i_ GDP which is tethered to the membrane. This forms the cortically localised NuMA/dynein-LGN- Gα_i_ complex. Dynein, due to its minus end-directed mobility, walks along astral microtubules towards the centrosomes. However, cortical localisation of the protein complex opposes dynein motion thereby exerting pulling forces on astral microtubules. These pulling forces then help to orient the mitotic spindle [74].

**Figure 7 jdb-10-00037-f007:**
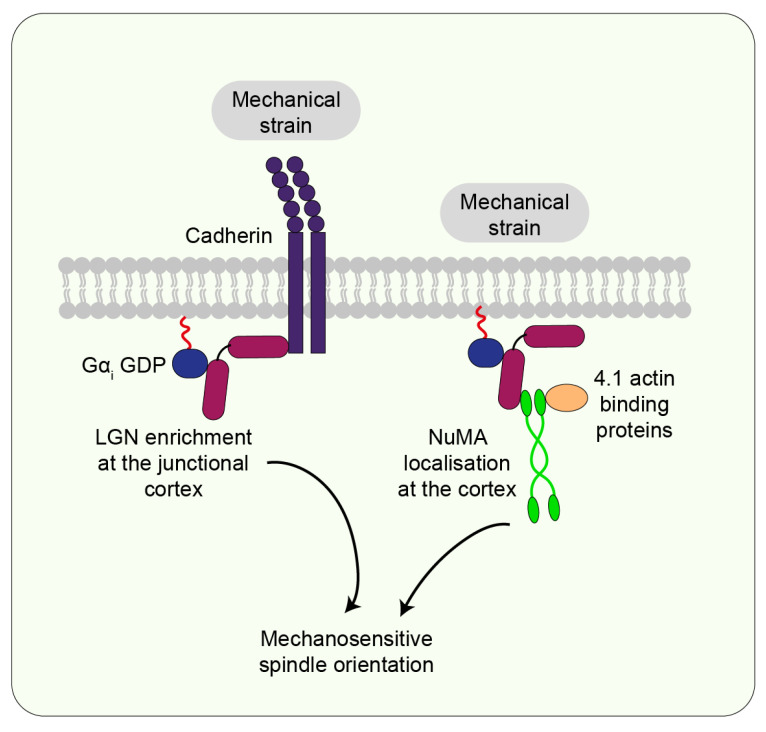
**Mechanical forces direct the localisation of key cortical proteins to influence spindle orientation.** Upon experiencing mechanical strain, cadherins promote the localisation of LGN to cell-cell contacts to regulate spindle orientation. Another spindle orientation core protein, NuMA, also localises to the cortex and potentially interacts with actin-binding proteins to influence mechanosensitive spindle positioning.

## Data Availability

Not applicable.

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
