# Peer review of "Sculpting an Embryo: The Interplay between Mechanical Force and Cell Division"

_jdb, 2022, doi:10.3390/jdb10030037_

Round 1

Reviewer 1 Report

Tarannum et al review the current understanding of the mechanisms by which mechanical forces regulate various aspects of cell division. The review is clear and comprehensive, discussing both underlying molecular mechanisms and their contribution to tissue morphogenesis. I have a few comments that might improve the manuscript.

1. In their discussion of myosin (starting with its role in E-cadherin mechanotransduction on p2 and continued on p11-p12) the authors seem to focus mostly on the role of myosin in linking forces to cellular responses. The role of myosin in actual force generation could be highlighted more. Most importantly, Figure 1 gives the impression that contractility of junctional actomyosin is only induced upon external forces, whereas actomyosin contractility can also be the origin of forces exerted on junctions that lead to cadherin mechanotransduction – which should be clarified. Similarly, in their discussion on myosin on p11-p12 it would help if the authors could make clear which of the described effects of myosin inhibition may be attributed to loss of force generation.

2. p3, line 95: the authors state that the position of the mitotic spindle at the end of metaphase determines the axis along which cells divide. It may be good to add some nuance to this, because there is significant evidence of mechanisms in anaphase/telophase that can direct spindle orientation and for instance correct misoriented metaphase spindles (e.g. see doi.org/10.7554/eLife.49249). 

3. The authors describe the role of Piezo and adhesion complexes in transducing forces to the cell cycle, but also the nucleus has been proposed to play a role in this (see for instance doi: 10.15252/embr.201948084). It may be good to briefly mention this in the manuscript. 

4. p5, line 146: when describing ref 43 (Streichan et al) the authors state that application of stretch increases entry into mitosis. This suggests regulation of G2/M progression (which is discussed later in this paragraph), whereas instead Streichan et al show stretch-induced cell cycle entry/progression of G0/G1 cells. It would be good to rephrase this sentence to clarify this. 

5. p6, line 170-174: In my opinion gut looping due to differential proliferation rates of the gut tube and underlying mesentery is not an example of differential mechanosensitive cell proliferation in organogenesis (as indicated by the authors). The differential proliferation rate is established by BMP gradients and (thus far) there is no evidence for mechanical signals in the regulation of proliferation rates. This should either be omitted or explained differently. 

6. In section 5.2, it would be good to include this recent publication: doi: 10.7554/eLife.78779. Furthermore, the discussion on ref 38 on page 11 (line 439-446) - explaining how mechanical stress can be sensed through cell shape changes but also without any contribution from cell shape - would be a good addition to mention in section 5.2.

7. p9, line 320-324: the discussion on mitotic rounding-induced pulling forces would benefit from some clarifications of the text. It would be more clear to specifically state that in ref 54 mitotic rounding was shown to exert pulling forces on the luminal surface of the epithelium (instead of only stating ‘pulling forces on the tissue’). More importantly, to my knowledge the description of the findings in the intestine are incorrect: mitotic rounding itself was not shown to induce compressive forces, but in compressed regions of the intestinal epithelium mitotic rounding causes the invagination (through apical-basal contraction of the mitotic cell).

8. In section 7, the authors describe the role of several mechanotransducers that link forces to mitosis. Although the separation of the text into paragraphs discussing the individual proteins makes the structure of this section more clear, it does feel a bit ‘fragmented’ and many of these proteins impinge on each other, which is now more difficult to understand when reading this section. A figure showing the role of these different proteins (including their connections) would therefore be very informative.

9. p10, line 372. Fat proteins, despite being part of the cadherin superfamily, are very distinct from the classical cadherins such as E-cadherin that have been introduced and discussed at length in the review. After discussing E-cadherin and when transitioning to Fat, I think it is important to emphasize this. 

10. Figure 2 is not referred to in the text. 

Author Response

We thank the reviewer for their time and for providing us with valuable suggestions. We have now modified the manuscript according to the reviewer’s suggestions for a better understanding of the article. All changes are indicated via track changes and respective line numbers.

  1. The caption for Figure 1 has been modified to clarify the role of myosin in force generation (lines 66-68). Similarly, in the discussion in section 7.5, whether the effects of myosin inhibition are due to loss of force generation has been clarified (line 516, 530-532).

  2. This sentence has been modified (lines 97-98).

  3. This has been incorporated into section 4 (lines 208-215). 

  4. This sentence has been rephrased (lines 162-164).

  5. This information has been explained differently (lines 188-193).

  6. The suggested paper has been discussed in section 5.2 (lines 339-342) as well as in section 7.3 (lines 473-476). The discussion on reference 43 (previously 38) has been incorporated in its entirety into section 5.2 (lines 349-356). We have additionally, removed this discussion from section 7.5 to avoid repetition and also because we believe it is a better fit in section 5.2.

  7. Both sentences have been amended (lines 374-375 and 376-378). 

  8. Figures 4 (section 7.1) and 6 (section 7.3) have been added for better clarity and understanding.

  9. This sentence has been modified (lines 443-445). 

  10. Figure 2 is now referred to in line 116.

Reviewer 2 Report

This is a very authoritative review of an important topic in developmental biology that is very timely. It will be welcome as a source of information and reference by both the developmental biology community and the cell biology community that is focused on cell division. The authors should be commended on the clear presentation and on the inclusion of examples from so many different model systems. There is a particular focus on Drosophila but that seems acceptable since so many different mechanisms have been investigated in this organism. One wonders, however, if the presence of a chitinous cuticle in some instances may affect the mechanics of cell division. While the article is quite good as is, the authors may want to consider the following suggestions, which in the opinion of this reviewer, would improve it even more.

1.    The figures provided are very well done, very informative. There are simply too few of them. This would be particularly helpful in section 5 where figures would aid a reader trying to sort through the relative contributions of cell shape and tension.

2.    The authors should consider a better balance of concepts. This is perhaps most evident in the different lengths of discussion given the concepts in section 5 and 6. It was surprising that mitotic rounding was discussed only to the extent that it was. Perhaps a bias of this reviewer, but this seems a particularly important concept that plays a key role in tissue morphogenesis and is well defined in particular tissues such as gut epithelium. Particularly surprising was the absence in section 6 of description of the work of Hu and Jasper in Drosphila gut epithelium. https://pubmed.ncbi.nlm.nih.gov/31509744/ This section would also benefit from a descriptive illustration.

3.    Finally, the one topic that received less attention than one would have thought and is of overriding importance in development is the mechanisms underlying asymmetric cell division. Asymetry in cell division is a critical concept in the existence maintenance of the stem cell niche. It is also critical and somewhat well studied in many early embryos, for example, C. elegans. Finally mechanisms have been investigated in some detail in Drosophila neuroblast division.

Author Response

We thank the reviewer for their time and for providing us with valuable suggestions. We have now modified the manuscript according to the reviewer’s suggestions for a better understanding of the article. All changes are indicated via track changes and respective line numbers.

In relation to most of the work having been done in Drosophila, we have added a few sentences (lines 494-500) to highlight the likely difference between invertebrate and vertebrate systems with respect to Mud/NuMA.

  1. Figure 3 has been added in section 5. We have also added Figure 4 (section 7.1) and Figure 6 (section 7.3).

  2. Since mitotic rounding has been a widely discussed topic elsewhere, we chose to focus on a few key examples in this review. We have, however, referred the reader to the review from Taubenberger et al (2020) which covers mitotic rounding specifically (line 371-373). We have also included two more examples of mitotic rounding in morphogenesis (lines 385-390).

  3. We agree that the asymmetric divisions are crucial when discussing development. However, very few studies have looked at mechanical forces, asymmetric cell divisions and development in unison. Since mechanical regulation of cell division is the focus of our review, we haven’t been able to incorporate much regarding asymmetric divisions. Therefore, the bulk of this review discusses the mechanical regulation of symmetric cell divisions. However, we have added some sentences clarifying this from line 134-140.

Reviewer 3 Report

The manuscript summarizes what is currently known about the interplay between mechanical forces and cell division. This is a very well-written review article and will be of broad interest to the community. The manuscript is almost in ready-to-publish form. I only have a few suggestions:

1. Since the manuscript is exploring ideas in the context of embryogenesis and tissue morphogenesis, it will be of broader interest to the developmental biology community if authors could summarize and explain how mechanical forces and cell division interact during two important stages of embryogenesis: gastrulation and body axis elongation.

2. The idea that cell divisions oriented in the same direction lead to tissue elongation (Figure2) is interesting, especially considering current progresses in embryo-like organoids which can develop either one axis or multiple axes after embedding into matrix. During both gastrulation (e.g. in Xenopus) and body axis elongation (e.g. chicken presomitic mesoderm elongation) in vivo, however, the elongation cannot be simply explained by oriented cell division as inhibition of cell division is either needed (e.g. Leise and Muller 2004) or ineffective (Bénazéraf et al, 2010) for elongation. The authors may want to clarify this.

Author Response

We thank the reviewer for their time and for providing us with valuable suggestions. We have now modified the manuscript according to the reviewer’s suggestions for a better understanding of the article. All changes are indicated via track changes and respective line numbers.

  1. Our review is structured such that discussions on gastrulation (lines 339-342, 387-390) and axis elongation (lines 359-367) are interspersed throughout. Since there is a wide range of model organisms as well, it would be challenging to bring everything under one umbrella in regards to gastrulation and axis elongation without completely restructuring the article.
  2. We have clarified these points in section 3 (lines 106-111). We have also highlighted an example where oriented divisions are crucial for tissue elongation (lines 106-108).